# Affordances for throwing: An uncontrolled manifold analysis

Timothy Bennett[1], Liam Thomas[1], Andrew D. Wilson[2]*

1 Carnegie School of Sport, Leeds Beckett University, Leeds, United Kingdom, 2 School of Humanities and Social Sciences, Leeds Beckett University, Leeds, United Kingdom

* a.d.wilson@leedsbeckett.ac.uk, DrAndrewDWilson@gmail.com

**Data Availability Statement:** All data and analysis files are available at https://osf.io/5aj3w/.

**Funding:** The authors received no specific funding for this work.

## Abstract

Movement systems are massively redundant, and there are always multiple movement solutions to any task demand; motor abundance. Movement consequently exhibits 'repetition without repetition', where movement outcomes are preserved but the kinematic details of the movement vary across repetitions. The uncontrolled manifold (UCM) concept is one of several methods that analyses movement variability with respect to task goals, to quantify repetition without repetition and test hypotheses about the control architecture producing a given abundant response to a task demand. However, like all these methods, UCM is under-constrained in how it decomposes a task and performance. In this paper, we propose and test a theoretical framework for constraining UCM analysis, specifically the perception of task-dynamical affordances. Participants threw tennis balls to hit a target set at 5m, 10m or 15m, and we performed UCM analysis on the shoulder-elbow-wrist joint angles with respect to variables derived from an affordance analysis of this task as well as more typical biomechanical variables. The affordance-based UCM analysis performed well, although data also showed thrower dynamics (effectivities) need to be accounted for as well. We discuss how the theoretical framework of affordances and affordance-based control can be connected to motor abundance methods in the future.

## Introduction

The human movement system is massively redundant: there are always many different ways of solving any particular movement task. This redundancy is a feature, not a bug; it enables our movements to be flexibly adapted to suit the specific current task demands (the 'bliss of motor abundance' [1]). However, it poses two related control problems at different scales [2]. The first is the problem of *action selection*: any given task has multiple movement solutions, so how do we select a good one? Formally, how do we select which action system variables to assemble into a functional system that solves the task? The second is the problem of *action control*; once we have selected a movement to execute, how do we implement the details? Formally, how do we set the correct parameters to the selected action system variables so the action unfolds appropriately?

**Competing interests:** The authors have declared that no competing interests exist.

A variety of mathematical methods have been developed to analyse the control of actions given the fact of motor abundance. These *motor abundance analyses* include nonlinear covariation analysis [3], tolerance-noise-covariation analysis (TNC) [4], stochastic optimal control theory [5,6], goal equivalent manifold analysis [7], and uncontrolled manifold analysis (UCM) [8,9]. The details vary, but each decomposes performance variability with respect to a *task goal*. This is because what counts as successful and stable action is only properly defined by that goal [10]. However, the methods vary in how they conceptualise what the task goal is. For example, TNC analysis relies on a dynamical analysis of the task-to-be-solved in an external frame of reference, while the other methods usually set the goal to be a biomechanical feature of the movement that seems most relevant to the success of the action, in an internal frame of reference. Exactly how the task is conceptualised affects the results of any analysis and the resulting hypotheses about movement control [11] and right now this remains a researcher degree of freedom rather than a principled, theoretically driven decision [6].

The goal of this paper is to investigate whether the *perception of affordances* can provide a meaningful framework to define what a task goal is; this paper contains a promised attempt to empirically demonstrate the concept [12]. Affordances are a key concept from Gibson's ecological approach to perception-action; they are perceptible action-relevant properties of organism-environment systems that define how an organism can act within a given task space [13–16]. They are generally studied as a solution to the problem of *action selection*, e.g., selecting either a stepping action or a climbing action depending on the height of a step measured with respect to the length of the individual's leg [17], or selecting optimal objects for throwing [18]. However, affordances have also been proposed to be a solution to the problem of *action control*, where they do not simply define which actions are functional options but also provide the ongoing basis for coordinating and controlling those actions [19]. In addition, affordances are real, definable parts of tasks that can be formally identified via a task dynamical analysis [12]. Affordances therefore have the potential to identify the task goal that we perceive in our task environments that meaningfully constrain action selection and control in the face of redundancy and motor abundance. In addition, they are the kinds of properties researchers can formally identify, and so they might also serve to usefully constrain research on action selection and control [10].

This paper is a first attempt to formally connect affordance perception and motor abundance analyses. In the following sections, we will briefly review UCM analysis, and a problem it has (specifically, how the selection of key variables for the analysis is under-constrained). We will then review the concept of affordances and how these may provide a solution to the problem, with a particular focus on the task dynamical affordance analysis of throwing developed in [20]. We then report the results of a throwing experiment designed to investigate whether the proposed affordance solution might work.

## Uncontrolled manifold analysis

Uncontrolled manifold (UCM) analysis begins with the fact that all movements are variable. Even if I attempt to produce the same outcome over and over again, and even if I succeed, I will never execute the action the exact same way every time. This 'repetition without repetition' [21] is possible because of the redundancy described above. The fact 'repetition without repetition' can be successful implies that not all movement variability is equal: there are a range of varied movement executions that still preserve the task outcome, and a range that do not. The action system seems to allow movements to vary within the space of possible executions that still preserve the outcome, while not allowing them to vary outside that space. This analysis of how movements are produced is formalised by the details of the UCM hypothesis and analysis [8,9].

In the language of UCM analysis, variables measuring the production of an action are called *elemental variables*, while variables measuring the task-relevant action outcome are called *performance variables*. Researchers must identify a system of elemental variables responsible for producing the performance variables (typically the time series of the joint angles involved in the action) and this system must be redundant (more elemental variable degrees of freedom than performance variable degrees of freedom). This system of elemental variables is then hypothesised to operate as a *synergy*. A synergy is a particular kind of dynamical organisation of degrees of freedom, such that perturbations to any of these are automatically compensated for by variation in the others so as to preserve the outcome of the synergy's activity. This produces 'repetition without repetition': variable movements with consistent outcomes. UCM analysis is a method for identifying whether a synergy is operating. It mathematically decomposes the total variance ($V_{TOT}$) in the elemental variables across repetitions with respect to the performance variables into two components. The first is variability in the elemental variables that does not prevent the system achieving the performance variables. This can be ignored by the action control system, as it is not going to hurt performance, and this subspace of variability lives on the *uncontrolled manifold*; $V_{UCM}$. The second component is variability that moves performance away from the performance variables. This must be controlled away to preserve a successful outcome, and this subspace of variability is *orthogonal* to the uncontrolled manifold; $V_{ORT}$. A synergy is present if there is an uncontrolled manifold present in the joint variability data. Specifically, if the ratio ($V_{UCM}$—$V_{ORT}$) / $V_{TOT}$ is greater than 0, the system of elemental variables is operating as a synergy to redundantly produce the performance variable for that analysis. This ratio is called the *index of synergy*.

UCM analysis therefore provides a way of testing various specific control hypotheses for a given task. A researcher can define a variety of candidate synergistic redundant systems (in the form of the selected elemental variables) and a variety of candidate task-relevant action variables (in the form of the selected performance variables) and look to see which pairings produces the largest indices of synergy. This provides insight into the actual control strategy (or strategies) being implemented by the person in the context of that task.

This process is under-constrained, however. For any given task, there are many elemental variables potentially involved in forming the synergy solving that task (although the important ones can usually be identified by considering the biomechanics of the action). It is harder, however, to identify which of the many potential task-relevant performance variables are at play. This is in part because even defining what the task is (from the first-person perspective of the organism) is a surprisingly hard thing to do correctly [22,23]. We will now turn to task-dynamically defined affordances and how they might help this problem.

## Affordances as dynamical properties of tasks

A performance variable is the thing the researcher hypothesises is being controlled by the action system. Ideally, the action system needs to be working to control a performance variable that suitably complements the task demands. To find the right performance variable, therefore, we need a clear definition of a task, and task relevance. This is not trivial [22,23].

In ecological psychology, task demands are formalised as *affordances* [13]. Affordances are typically considered as dynamical properties of organism-environment systems; properties of the environment measured with respect to properties of the organism [14,15]. Dynamics is the mathematics of how systems change over time and the forces involved in those changes, and it provides a complete framework for formally describing the ecological-scale, organism-environment system.

This ecological task-dynamical analysis [22,24,25] has been specifically applied to the task of targeted long-distance throwing [12,20,26], and this illustrates how it leads to a formal

description of task affordances. Dynamically, throwing is an example of projectile motion, where the projectile is briefly propelled but then left to fly without further control by the thrower. Any projectile motion can be fully described by an equation of motion that contains information about the projectile (specifically size and mass), about the conditions at release (release angle, release speed, and release height), and about some environmental factors (specifically air density, drag, and gravity). A successful targeted throw is one in which the variables are all parameterised such that they result in a motion that intercepts the target (so the full dynamical description includes a specification of the target as well; distance, size, and orientation). Setting those parameters is the goal of the dynamics of the throwing action; the task dynamics define the task for the action system, and, we propose, can therefore also meaningfully define the task for the researcher applying a motor abundance analysis.

This task analysis for targeted throwing was developed and tested [20]. They used a single constant object (a regulation tennis ball) which kept the projectile and environmental factors constant, and a single constant target (a Perspex square 1.2m on each side), and then varied the target location in terms of height (target centre at 1m, 1.5m and 2m) and distance (5m, 10m, 15m). They then measured how release angle, speed, and height varied when skilled throwers hit the target with the tennis ball, and showed that all throwers scaled their actions to the different target locations. Note how all the experimental measures and controls come from the task dynamics, specifically the equation for projectile motion.

The next step [20] used that equation to generate simulations, not just of the release parameter combinations that were produced, but of the full range of release parameters that were on offer. Each simulated throw produced a hit or a miss, and each target location resulted in a slightly different set of release parameter combinations that produced hits. Plots of these sets mapped *the affordances of the target to be hit by a projectile motion of a tennis ball*. They then related the release parameters produced by people to the set on offer by each target, and showed that performance mapped neatly onto the task dynamically derived affordances. (This affordance analysis connects to another motor abundance analysis, specifically tolerance-noise-covariance (TNC) analysis [4]; we will return to this topic in the Discussion).

Wilson et al [20] used a task dynamical affordance analysis to explain the outcomes of throwing dynamics (the selected release parameters). This paper will take the next step and use the affordance analysis to explicitly inform an UCM analysis of those throwing dynamics (the control of the action that produces those selected release parameters). Ecologically, these throwing dynamics are referred to as *effectivities*, and they are the dynamical properties of the organism that are complementary to the affordance properties of the environment [27].

## Uncontrolled manifold analysis of throwing

There have only been a few papers applying UCM analysis to targeted ball throwing. They all evaluate variability in the shoulder-elbow-wrist system (sometimes including the fingers as well) as the elemental variables. Interestingly, they use a wide variety of tasks and performance variables (which nicely illustrates how under-constrained the selection process is) and consequently have found a variety of results.

Two papers have examined dart throwing as their task. The first [28] used hand position defined with respect to the shoulder as their performance variable, and found the index of synergy was negative throughout the throw (no synergy present in the joint angle data relative to hand position). The second [29] used hand position (mean hand path in an external frame of reference) and hand centre of mass (internal frame of reference) as their performance variables; they found that the index of synergy for both was only >0 late in the throw. (A third

paper looked at this task, but it was unclear from the paper which elemental variables were used in which analysis [30].)

Another paper examined basketball free throws, and examined six performance variables [31]. They looked at hand and finger position each in both the longitudinal axis and the vertical axis, and hand and finger orientation as well. The index of synergy was >0 for all variables, but the two position along the y axis variables performed the best. (This paper also used TNC analysis [4] on the ball release parameters as a first attempt to look at both redundancy in the body (via UCM) and ball (via TNC) to see how they relate.)

A final paper [32] had participants throw a small ball into a basket and used hand position and velocity (measured in an external frame of reference) as their performance variables. The index of synergy for both was >0 across the movement, consistently around 1 for position but starting high and steadily decreasing over the movement for velocity.

These papers demonstrate a few key points. First, performance is highly task specific and even these examples of 'throwing' are sufficiently different that the control mechanism is different in each case; a detailed task specification is required. Second, different authors used different performance variables measured in both internal and external frames of reference, with a variety of justifications; again, a detailed task specification is required to nail down what is actually important for task success. Third, even the selection of elemental variables was varied. Some included the fingers, and one [30] included lower body joint angles in one of their analysis. This makes sense given that shoulder-elbow-wrist systems don't simply float in the air as they produce throws, but whether these variables are actually critical to the throw remains unclear. Fourth, reporting was highly variable; the variable selection process was not always explicitly justified and establishing the elemental variables, performance variables, and frame of reference was not always easy, and these all have consequences for interpreting the results [11].

### The current study

The goal of this paper is to build on existing throwing and UCM research and to demonstrate that an ecological, task-dynamical affordance analysis can serve to guide and constrain both experimental design and motor abundance data analysis. We therefore replicated the throwing task from [20] for which we had the affordance analysis. We focused on the distance manipulation from that study because altering the distance had the most effect on movement execution (all release parameters systematically varied as a function of distance). We analysed variability in the shoulder-elbow-wrist elemental variable system using UCM analysis to match the studies above. We did this UCM analysis with respect to nine different performance variables, six biomechanical ones (hand position, velocity, and orientation, each defined either with respect to the shoulder or with respect to the lab) and three derived from the affordance analysis (release angle, release speed, release height, all defined with respect to the lab). We aimed to see whether the affordance performance variables produced high indexes of synergy, and to show how they performed compared to the biomechanical performance variables.

### Methods

All anonymised data and analysis files are available at https://osf.io/5aj3w/; the study was not pre-registered.

### Participants

10 experienced male throwers participated in this project and were injury free during the period of testing (April, 2017). Three participants were excluded from analysis due to substantial marker dropout on multiple trials, leaving 7 for the final analysis. Each participant gave

written informed consent and the research project received ethical approval from the Carnegie Faculty Research Ethics Committee of Leeds Beckett University. Note that as we did record high speed video, individual participants could be identified; however these videos are not shared and are stored securely on University cloud systems.

## Design

There was one within-subjects factor, Target Distance, with 3 levels (5m, 10m, 15m). We measured the three release parameters as dependent variables; Release Angle (˚), Release Speed ($ms^{-1}$), and Release Height (m), and analysed these using a repeated-measures ANOVA in JASP 0.16.0.0 to replicate the basic result from [20]. The assumption of sphericity was violated for all analyses, so we report the Greenhouse-Geisser corrected p values.

For the UCM analysis, we required a set of elemental variables (joint angles to be controlled), and performance variables (candidate variables that define how the elemental variables are being controlled). We will describe the details of these measurements and calculations in the UCM analysis section below. No statistical analysis was conducted on these data, however.

## Apparatus

Three-dimensional kinematic data was captured at 250Hz using ten infrared cameras (Oqus 700); 6 attached to an overhead rig and 4 floor-based cameras located around the calibrated capture volume. The cameras were connected to the Qualysis Track Manager (QTM) motion analysis system (Qualysis AB, Gothenburg, Sweden).

Using a calibrated anatomical system technique [33] fifteen retro-reflective markers were used to track the throwing arm for both static and dynamic motion trials. The markers were attached to the following locations: incisura jugularis (IJ) and 7th cervical vertebrae (C7), xiphoid process (XP), 8th thoracic vertebrae (T8), left and right upper back markers (tracking) left and right anterior, central-medial and posterior acromion, markers (i.e. acromion triad), three non-collinear markers placed on the central region of the upper arm (tracking), medial and lateral epicondyles, non-collinear markers placed on the central region of the lower arm, radial and ulnar styloid process, second and fifth head of metacarpal. All body markers were 12.5 mm in size.

A 1000Hz floor mounted force platform (Kistler, Model Switzerland) was used to define the start of the throwing motion (once the vertical ground reaction force exceeded a 10N threshold) and located centrally in line with the target. One high-speed camera was positioned perpendicular to the force platform at a distance of 5m to measure ball release, and a second was positioned behind the target to measure impact location (both Fastec TS3, recording at 250Hz).

All measurement systems were synchronised by use of a single trigger that began recording on all systems simultaneously.

## Procedure

Testing took place at the Carnegie Research Institute at Leeds Beckett University. Participants performed self-directed warm-up prior to testing. The participants were then allowed to familiarise themselves with the throwing task with all markers attached for both static and dynamic trials.

Participants threw a regulation size and weighted tennis ball at a vertically orientated target (1.2m x 1.2m in size, with a 30cm diameter circle and crosshairs marked in tape with the centre of the crosshairs which was located 1.5m from the floor) located at one of three distances (5m,

10m and 15m) from the centre of the force platform. Distance was randomized across the session and researchers moved the target to the correct location for each trial. Participants were instructed to take a self-selected one-step approach onto the force platform using their stride leg; this constrained participants into a fairly standardised throwing action. Participants were also instructed to aim for the centre of the target, with an emphasis placed on throwing for accuracy. All participants threw the ball with their dominant hand (6 right- and 1 left-handed). 20 throws that successfully hit the target for each distance were recorded for further analysis; misses were replaced at the end of the randomized session order until 20 hits had been achieved for each distance condition (details in 'Trial Order and Record.xlsx' on the OSF https://osf.io/5aj3w/).

## Data analysis

We did two separate analyses on the data. First, we examined the release parameters to test whether we had replicated the basic effects from [20]. Second, we examined the kinematics of the shoulder-elbow-wrist system during the execution of the throw using UCM analysis. This was a very exploratory analysis approach, in which we decomposed the movement variability in the 7DOF shoulder-elbow-wrist system with respect to the nine different candidate performance variables, to see which, if any, identified the highest level of synergy variability.

## Release parameters

For each trial, the time of ball release was identified from the video taken with the 'release' camera. The 2D ball trajectory (x,y image coordinates) was digitised for 10 frames on either side of the release frame, and these were converted to lab coordinates according to a calibration scale. These time series was low pass filtered (frequency = 10Hz) and release parameters were calculated using these data.

- *Release Speed*: the filtered x and y coordinate time series were differentiated to calculate the speed in each component $V_x$ and $V_y$, and the resultant velocity $V_{xy}$ was computed from these. Release speed is defined as $V_{xy}$ of the ball in the release frame.

- *Release Angle*: this was computed as $\tan^{-1}(V_x, V_y)$ in the release frame.

- *Release Height*: this was directly measured as the filtered y coordinate of the ball in the release frame.

## Throwing kinematics

The overhand throwing technique was measured from initial ground contact of the stride leg to ball release. Ball release was identified from the 'release' camera as described above. Because all the measurements were synced, these time stamps could also be used on the data from the Qualisys cameras.

Once the start and end times were determined, a cubic spline interpolation was used to normalise all movement trajectories to 100% (100 samples) using Visual3D software (Version 6 Professional C-Motion, USA) to align each trial for further analysis. Residual analysis was used to determine the appropriate cut-off frequency for all segments using a fourth order Butterworth low-pass filter. A cut-off frequency of 10 Hz was applied to all segments of the throwing arm.

*Joint angles and segment calculations.* A 4-segment forwards kinematic model for the throwing arm was created using Inverse Kinematics (Visual 3D version 6 Professional, C-Motion, USA) allowing 3-D rotation at the shoulder (abduction-adduction, flexion-

extension, internal-external rotation) and elbow (flexion-extension, forearm supination-pronation) and wrist joints (flexion-extension, abduction-adduction). Local coordinate system for the thorax, upper arm, lower arm, and hand were based on ISB recommendations [34] with the positive z-axis aligned vertically upward, positive y-axis directed from posterior to anterior, and positive x-axis pointed to the right in a medial to lateral direction. The shoulder was calculated relative to the thorax using a ZYZ Euler sequence, while the elbow and wrist were calculated relative to the proximal segment using an XYZ Cardance sequence [34].

## Uncontrolled Manifold (UCM) analysis

Data and analysis code in the 'UCM Analysis' folder on the OSF https://osf.io/5aj3w/.

The UCM method tests whether trial-to-trial variability of elemental variables (e.g. joint angles) are structured to stabilise performance variable(s) that are potentially important for the successful completion of a given task.

*Elemental Variables*: We measured the time series of joint angles for the shoulder-elbow-wrist system executing the throw. The shoulder and wrist both have 3 degrees of freedom, while the elbow has 1, for a total of 7 degrees of freedom. We measured this for each of 20 successful throws to each of the three Distances.

*Performance Variables*: we investigated a total of nine candidate performance variables. Three related to the hand (hand orientation, hand position, hand velocity), and were measured in either an Internal (with respect to the shoulder) or External (with respect to the lab) frame of reference. Each of these has 3 degrees of freedom. Three related to the affordance analysis (release angle, release speed, and release height) and were measured in the External frame of reference (with respect to the lab). Each of these has 1 degree of freedom. In all cases $DF_{EV} > DF_{PV}$, defining a redundant system.

A Jacobian (J) matrix determines how small changes in the elemental variables alter the value of a performance variable. The Jacobian was estimated across all trials (j) at each percentage (i) of the normalised whole movement trajectory (i.e. from ground contact to ball release) using the coefficients (k) of multiple linear regression between mean-free joint angles ($\underline{\theta}_{ij} - \underline{\theta}^0_i$) of the wrist ($\theta^{1i}, \theta^{2i}, \theta^{3i}$), elbow ($\theta^{4i}$), and shoulder ($\theta^{5i}, \theta^{6i}, \theta^{7i}$), and changes in the mean-free instantaneous value of the performance variable [35].

The UCM was approximated by the null-space, a linear subspace of the Jacobian within which variations of joint configurations keep the performance variable unchanged. The null-space here is spanned by (n-d) basis vectors, $\varepsilon_i$ and is expressed as:

$$0 = \underline{J}(\underline{\theta}^0) \cdot \varepsilon_i \tag{1}$$

Deviations of the joint configurations from their mean ($\underline{\theta} - \underline{\theta}^0$) were then projected onto the null space and the component perpendicular to the null-space using Eqs 2 and 3 respectively:

$$\underline{\theta}_\parallel = \sum_{i=1}^n \varepsilon \cdot (\underline{\theta} - \underline{\theta}^0) \tag{2}$$

$$\underline{\theta}_\perp = (\underline{\theta} - \underline{\theta}^0) - \underline{\theta}_\parallel \tag{3}$$

The amount of variability per DOF within the UCM (i.e. $V_{UCM}$) was estimated as:

$$\sigma^2_\parallel = (n - d)^{-1} \cdot (N_{trials})^{-1} \cdot \sum \theta^2_\parallel \tag{4}$$

where n is the number of degrees of freedom in the elemental variables, d is the number of degrees of freedom in the performance variable and $\theta_{ll}{}^2$ is the squared length of the deviation vector $\theta_{ll}$ lying within the linear UCM [8]. The amount of variability per degree of freedom orthogonal to the UCM (i.e. $V_{ORT}$) was estimated as:

$$\sigma_{\perp}^2 = d^{-1} \cdot (N_{trials})^{-1} \cdot \sum \underline{\theta} \perp^2 \qquad (5)$$

The index of synergy for each performance variable was calculated as the difference between $V_{UCM}$ and $V_{ORT}$ scaled by the total variability ($V_{TOT}$) [36,37]

$$\Delta V = \frac{V_{UCM} - V_{ORT}}{V_{TOT}} \qquad (6)$$

where

$$V_{TOT} = \left(\frac{1}{n}\right)\left(\text{d } V_{ORT} + (n - d)V_{(UCM)}\right) \qquad (7)$$

Positive values for $\Delta V$ (i.e. $V_{UCM} > V_{ORT}$, so $\Delta V > 0$) supports the existence of a synergy. Negative values for $\Delta V$ (i.e. $V_{UCM} < V_{ORT}$, so $\Delta V < 0$) does not support the existence of a synergy [37]. A higher $\Delta V$ suggests a stronger synergy, although this outcome is dependent on the magnitude of $V_{ORT}$ (because $V_{ORT}$ directly affects the value of the performance variable) [37,38].

## Results

### Release parameter analysis

Data and analysis code in the 'Ball Release Data' folder on the OSF https://osf.io/5aj3w/, and refer to Fig 1.

For each participant, we computed the mean release speed, release angle, and release heights across trials, and ran a repeated measures ANOVA for each dependent variable, with Distance (3 levels: 5m, 10m, 15m) as the only factor.

- *Release Speed*: there was a significant main effect of Distance, $F(1.08, 7.57) = 5.5$, $p < .05$, partial $\eta^2 = 0.44$, where speed increased linearly with distance.

- *Release Angle*: there was a significant main effect of Distance, $F(1.03, 7.23) = 41.1$, $p < .01$, partial $\eta^2 = 0.86$, where angle increased linearly with distance.

- *Release Height*: there was a significant main effect of Distance, $F(1.14, 7.98) = 15.3$, $p < .01$, partial $\eta^2 = 0.69$, where height increased linearly with distance.

This replicates the effect of distance from [20]; participants were indeed scaling their throws to suit the distances.

### UCM analysis

We decomposed joint angle variability in the shoulder-elbow-wrist system with respect to the nine different performance variables. For each performance variable at each Target Distance, we computed the total variance $V_{TOT}$, and the two components $V_{UCM}$, and $V_{ORT}$. We then computed the index of synergy, $\Delta V$ across the whole normalised time window in the throwing motion defined as the segment between initial ground contact of the stride leg to ball release.

Three of the performance variables were defined in an Internal Frame of Reference, specifically with respect to the shoulder. These were the Orientation of the hand, the Position of the

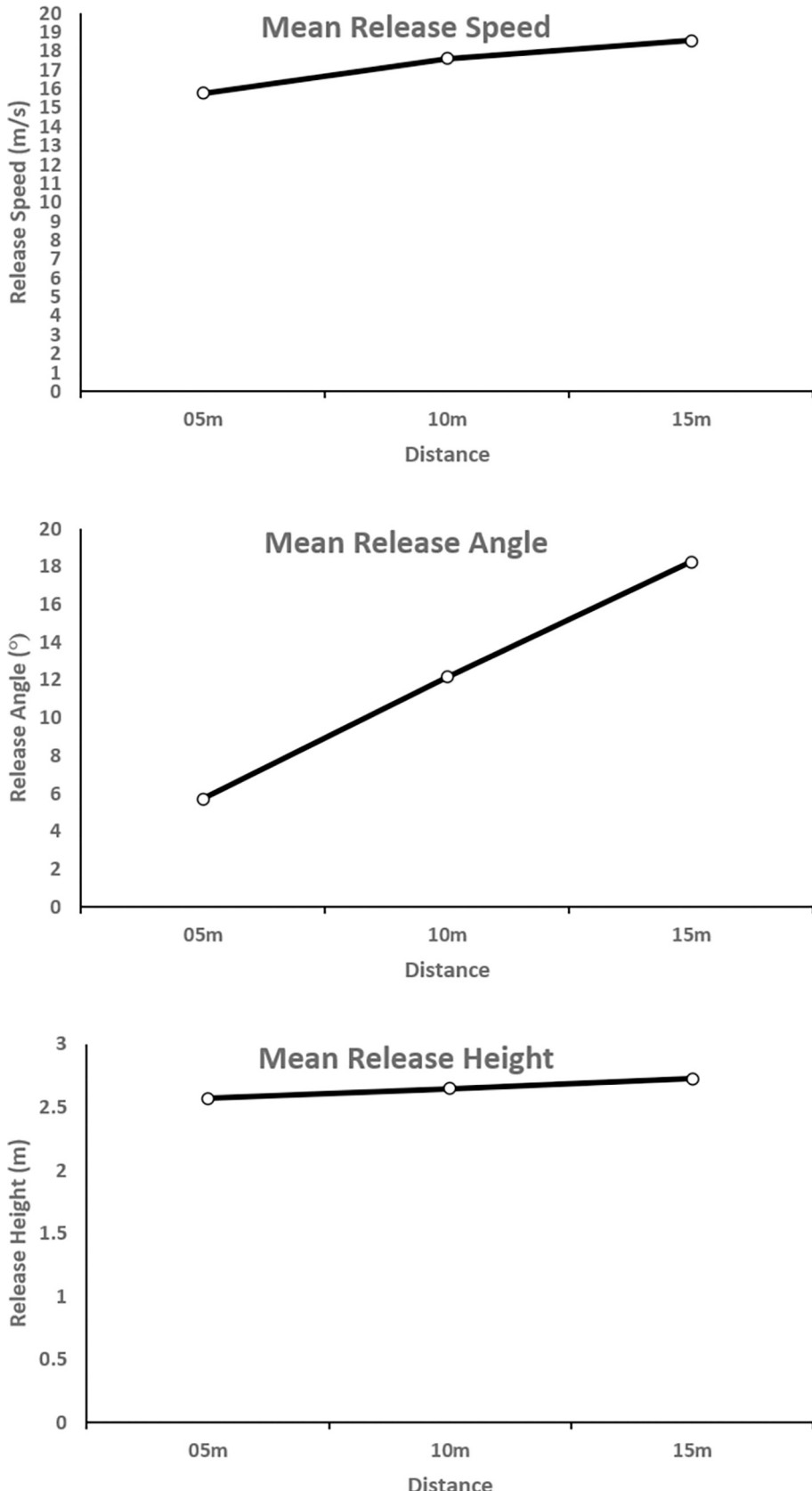

**Fig 1. Release parameters.** Release parameters produced by participants plotted as a function of Distance.

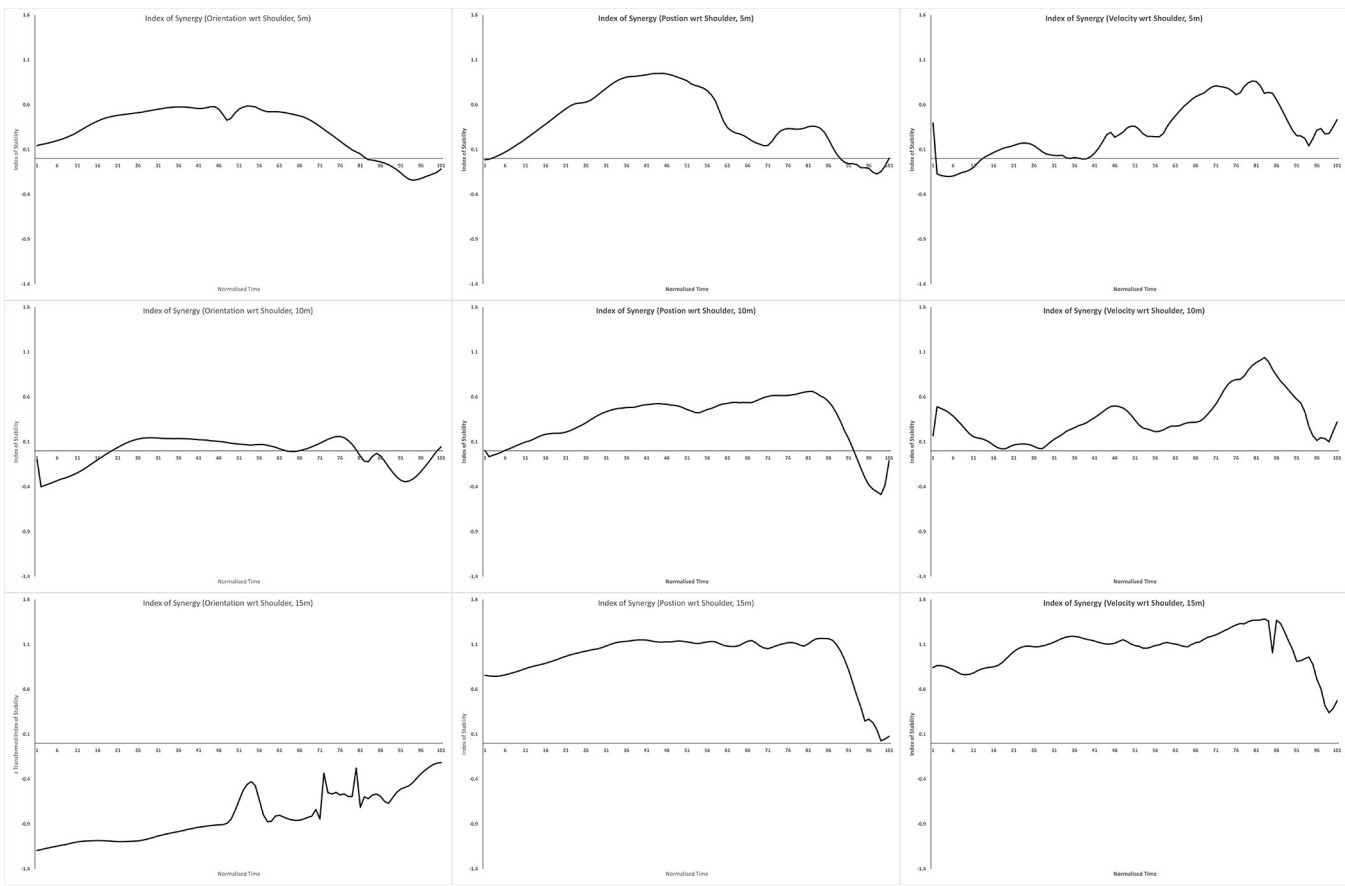

**Fig 2. Internal frame of reference.** Plots of the mean index of synergy across the throw for hand orientation (left column), hand position (middle column) and hand velocity (right column), defined with respect to the shoulder (internal frame of reference). The top row plots data for throws to 5m, the middle row, 10m, and the bottom row, 15m.

hand, and the Velocity of the hand. Three of the performance variables were defined in an External Frame of Reference, specifically with respect to the lab. These were (again) the Orientation of the hand, the Position of the hand, and the Velocity of the hand. Finally, we used the three Release Parameters as performance variables, Release Angle, Release Speed, and Release Height. These were also measured in an External Frame of Reference, specifically with respect to the lab.

We computed $V_{UCM}$ & $V_{ORT}$ for each participant, and calculated the average of each across participants. We then computed the Index of Synergy for each performance variable at each distance using these averages; these are plotted in Figs 2–4. A positive index of synergy indicates the presence of a synergy in the elemental variance variability with respect to that performance variable.

To summarise each plot, we computed the proportion of time each index plot was >0.5. The throwing action we were measuring (from initial ground contact of the stride leg to ball release) consists of a single basic phase, and this proportion measure tells us how consistently across this phase a synergy was present. This kind of proportion measure has been used to meaningfully summarise these kinds of time series in research on coordinated rhythmic movement [39]. We measured the proportion > 0.5 in order to impose some notion of 'meaningfully greater than 0' (this is another researcher degree of freedom in UCM analyses, and there

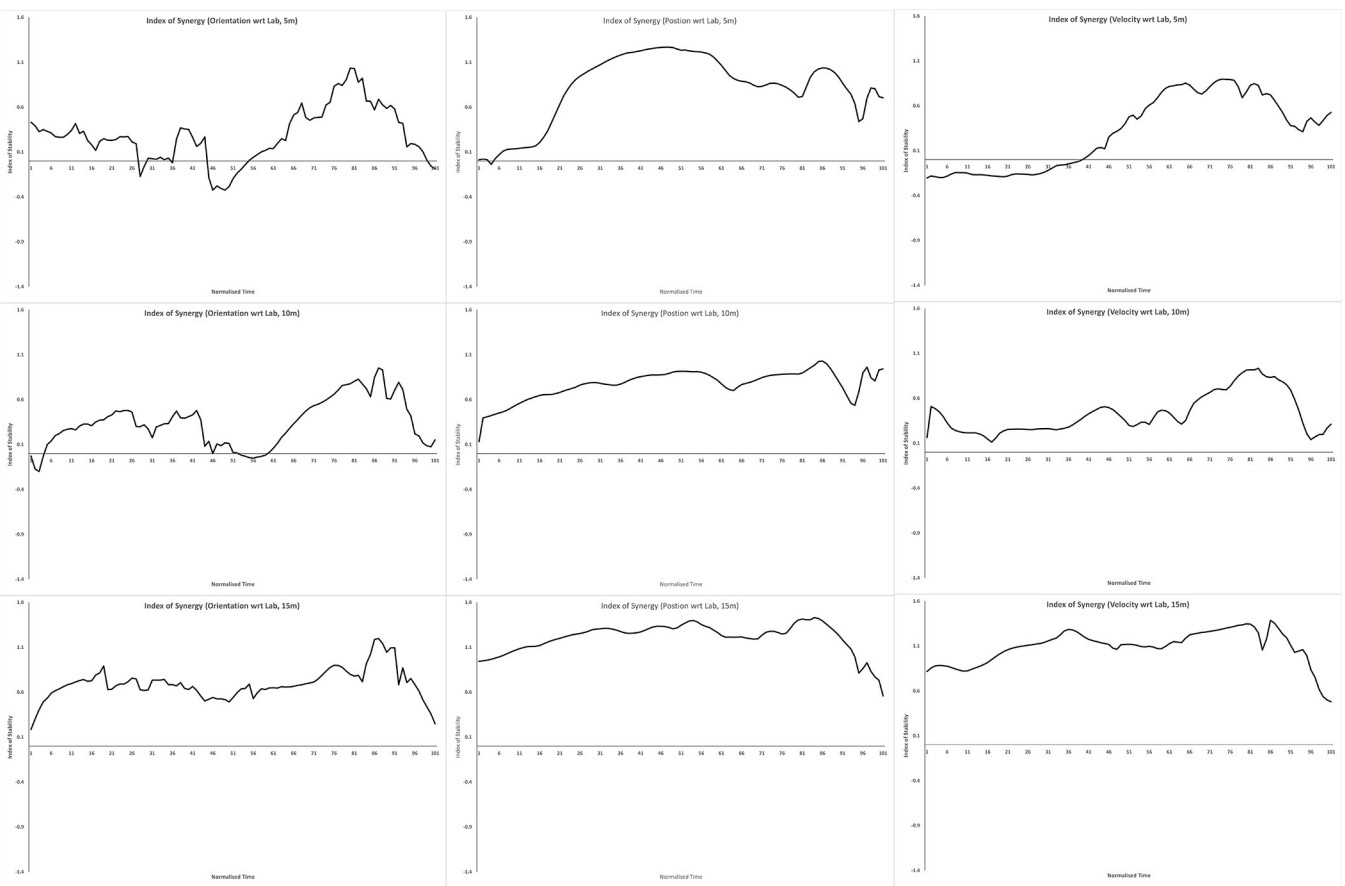

**Fig 3. External frame of reference.** Plots of the mean index of synergy across the throw for hand orientation (left column), hand position (middle column) and hand velocity (right column), defined with respect to the lab (external frame of reference). The top row plots data for throws to 5m, the middle row, 10m, and the bottom row, 15m.

are no set guidelines here. We chose 0.5 here based on visual inspection of the data, but we suggest it would be worth developing some justified criteria here). The proportions are plotted in Fig 5.

## Discussion

This paper was a first attempt to use an ecological, task-dynamical affordance analysis to inform an UCM analysis of action control variability. While the affordance performance variables did well, the overall pattern was a little complicated and there are several things to note from the UCM.

First, performance variables defined with respect to the shoulder (Internal Frame of Reference; mean across all performance variables and distances 0.39) perform worse than those defined with respect to the lab (External Frame of Reference, mean 0.63, Release Parameters, mean 0.57). The throw is being organised with respect to external task variables (which aligns with the affordance hypothesis).

Which variables mattered was different at different distances, however:

- Each distance produced a very different profile across the candidate performance variables. At 5m, Position with respect to the Lab and the release parameters dominated; at 10m, only

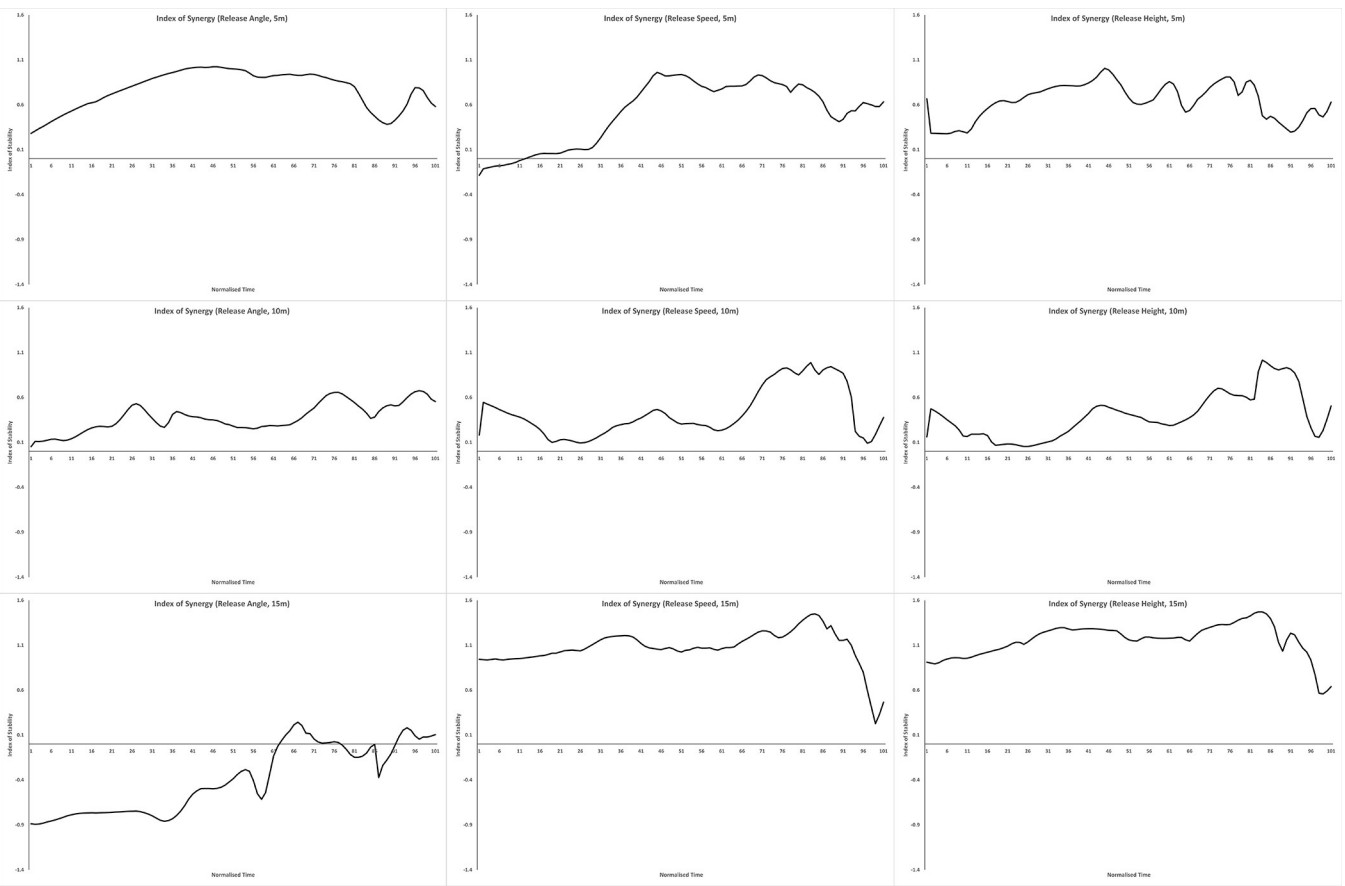

**Fig 4. Affordance analysis.** Plots of the mean index of synergy across the throw for release angle (left column), release speed (middle column) and release height (right column), defined with respect to the lab (external frame of reference). The top row plots data for throws to 5m, the middle row, 10m, and the bottom row, 15m.

Position with respect to the Lab mattered; at 15m, everything except Orientation with respect to the Shoulder and Release Angle mattered.

- The only performance variable that scored consistently high was Position with respect to the Lab (mean across distances of 0.9). This makes sense; a stable trajectory is what enables a smooth acceleration with minimal force lost as the kinetic chain unfolds down the arm, and this is true for all distances.

- All three release parameters from the affordance analysis performed well at 5m, and release angle height and speed performed well at 15m.

- The variation with distance was not linear; it was not simply the case that various performance variables became increasingly important as the distance increased. In fact, 10m seems to be the easiest distance; note that at 10m only Position with respect to the Lab defined a synergy, and also see Fig 6 which plots the average total variance $V_{TOT}$ for the three distances; throwing to 10m was the least variable. If you order the Proportion Index of Synergy $>0.5$ plots in the order from the t. otal variability (10m, then 05m, then 15m), there is more of a linear progression; as the throwing task gets harder, more performance variables become relevant and become more relevant.

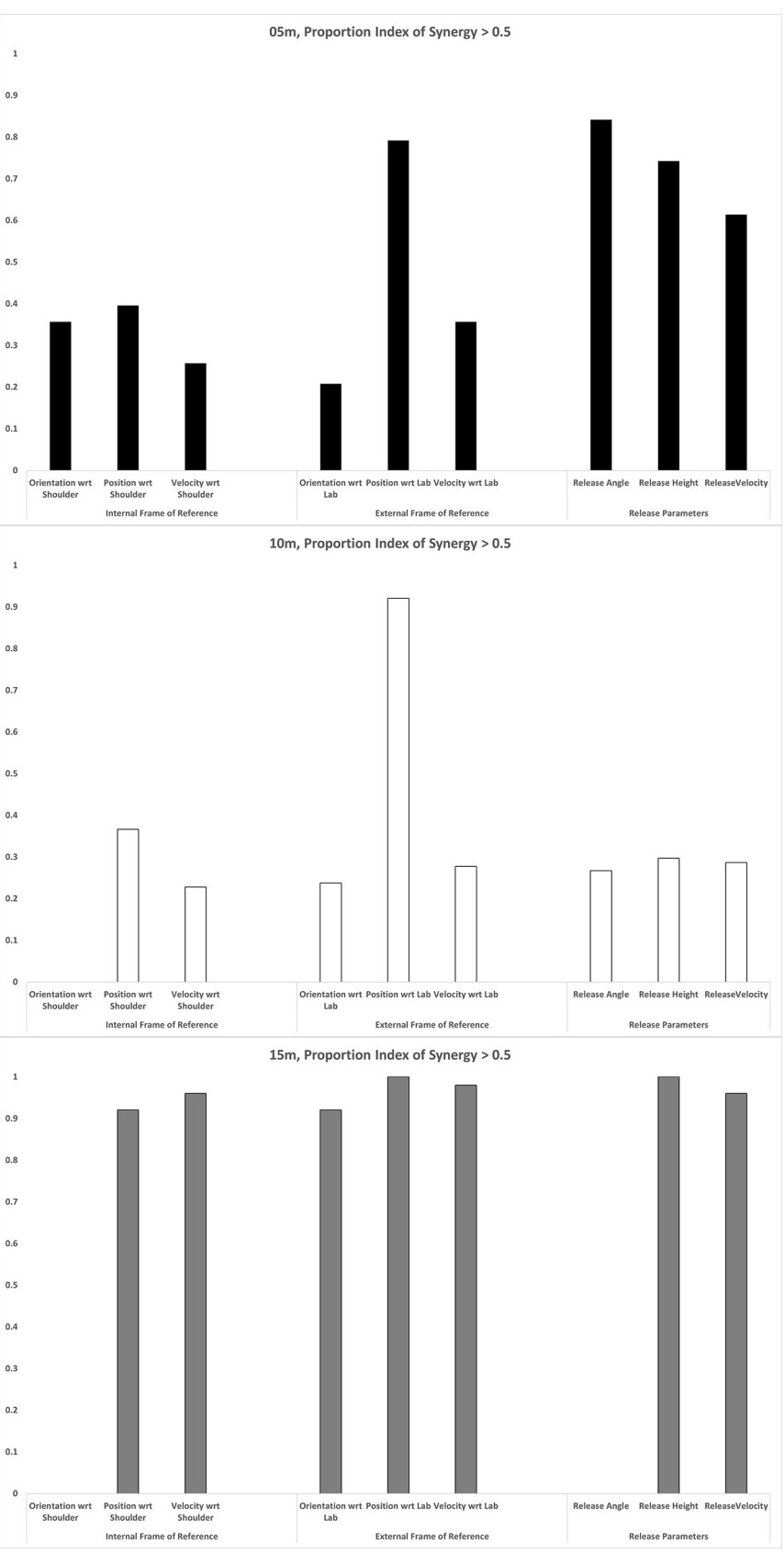

**Fig 5. Proportion plots.** The proportion of each mean index of synergy time series that was >0.5 for each of the nine performance variables and for each distance (top, 5m; middle, 10m; bottom, 15m).

Broadly, then, using an affordance analysis to inform the UCM was a success, but it is, as yet, an incomplete account. In the next section, we will review a few of the lessons learned from this study and how they relate to a variety of other projects in the field of motor control.

## Action selection and action control

Affordances are generally discussed and studied as a solution to the problem of action selection. This is even true of the affordance analysis we rely on here [20]. In that paper, the simulated affordance maps were proposed as a formal description of the affordance property guiding the selection of release parameters. We equated the affordance to the set of release parameters that would result in hitting a target at a given location and showed that the structure of that set shaped the variability in the selected release parameters.

Treating affordances as the constraint on action selection in this way more naturally aligns with the tolerance-noise-covariance method (TNC [4]). TNC is literally a method for decomposing the variance in selected movement parameters relative to a task-dynamical analysis. But this alignment implies a clash with methods such as UCM. Sternad and colleagues [11] identified how the coordinate frame in which action data are analysed in is a researcher degree-of-freedom that has major consequences for the results and therefore the interpretation of the data. They specifically use UCM as an example of an analysis that is susceptible to this problem and advocate instead for the use of a task-dynamical analysis to provide the necessary constraint on research on redundant action systems; TNC, and not UCM.

We agree with Sternad [11] about this issue: UCM analysis is indeed under-constrained, and this has consequences for how useful it is as a method for examining control processes. It is under-constrained in two ways: as Sternad et al note, the coordinate frame of reference for

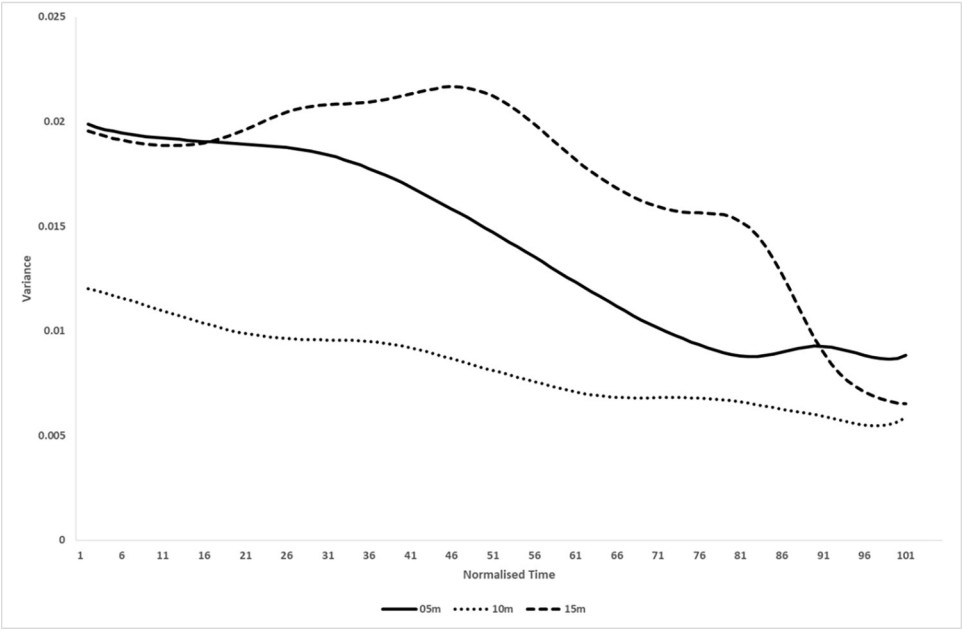

**Fig 6. Total variance.** Plot of the mean total variance for each distance.

the elemental variables is a researcher degree of freedom and which one is used produces very different results. Then, as we discussed in the Introduction, the selection of performance variables is also a researcher degree of freedom and the selection is rarely formally justified (certainly the throwing papers we reviewed did not do much to justify their selections, which were variable across papers). However, rather than simply not use methods such as UCM, we advocate two related solutions.

First, we advocate treating the variable selection freedom as a feature, not a bug, by more explicitly engaging with it and its consequences. UCM was proposed as a method for testing hypotheses about motor control processes, where researchers look to see which of a set of candidate performance variables are, in fact, constraining the synergy producing the behaviour of a set of elemental variables. Being systematic here will allow UCM analysis to rule different control hypotheses both in and out. In reviewing papers applying UCM to throwing, it was notable that the selection process for elemental and performance variables was rarely discussed and never formally defended (to the point where it could be quite difficult to establish what the variables even were). More care will enable UCM research on various tasks to build and develop more systematically.

Second, however, is acknowledging the fact that UCM by itself does not provide the necessary constraints on variable selection. While it does embody a particular understanding of the problem of action selection and control, it is fundamentally only a method, not a theory. To support systematic research, it needs to be applied to test hypotheses generated by theories of perception-action mechanisms. The ecological approach is such a theory [40] and it can provide candidate mechanisms [41,42]. We worked here to show that a task-dynamical affordance analysis can provide a meaningful constraint to guide the systematic selection of candidate performance variables. In effect, we wanted to show that affordances can help us understand not just action selection, but action control, and our results showed promise when compared to more typical biomechanical variables.

It is not a complete solution, however: for example, one key performance variable across all distances was hand position defined with respect to the lab, and this variable does not feature in the affordance analysis. However, this result provides a link to other work on using affordances to understand action control, which we can now discuss.

## Affordance based control

Fajen introduced the notion of affordance-based control, in which he proposed that the real time control of actions comes from the calibrated perception of affordances [19]. This calibration is the critical piece we have not yet included in our analysis, and it reflects the contribution of participant dynamics to the perception and use of affordances.

Affordances offer a wide range of options to an organism. The affordances of the target to be hit by a tennis ball, for example, is a large set of release parameter combinations. Not all of these are available to every organism, however. For example, any of the targets can be intercepted with a low release angle and a release speed of $60ms^{-1}$, but given that human throwing tops out around $45ms^{-1}$ that option isn't available; and for a less skilled thrower, even possible speeds such as $35ms^{-1}$ may be out of reach. Fajen [19] called this an action boundary, and noted that actual movement must work to respect these boundaries; if the affordance on offer is on the wrong side of that boundary then the action is not possible for that individual at this time and a new response is required. These boundaries are defined by organism effectivities, which are the dynamical properties of the organism that complement a given affordance [27], and these effectivity properties provide the action-relevant scaling that calibrates the perception of affordances so that it can guide actual action selection and control in the moment.

Simple effectivity measures have been used in affordance research since the original Warren paper; he had people of different heights judge the climbability of steps and showed that this organism property scaled the affordance perception [17]. More recent work has shown that the relevant effectivity properties are dynamical rather than geometric; how movable the leg is vs simply how long it is [e.g. 43,44]. Fajen [19,45–47] uses visually guided braking as his task which enables him to manipulate the effectivity dynamics by altering the dynamics of the brake; people can perceive these changing dynamics and use that perception to scale their use of the braking affordances in a task to keep a safe deceleration an option. This calibration process itself can also be studied [e.g. 48]. Connecting this research to affordance-constrained UCM analysis will be a valuable development.

### Reflections on using UCM for throwing

There are some caveats we would like to mention in applying UCM the way we have to a large whole-body motion such as throwing, as points to consider in future research.

### Which elemental variables?

First, throwing *is* a whole-body motion, and we have only analysed the variance in the shoulder-elbow-wrist system. As noted in the Introduction, this is standard in the existing papers, although it is never explicitly justified, and it may be the case that a more whole-body approach is required. That said, while throwing is a continuous action, it can be analysed into 6 distinct biomechanical phases. The lower body and torso make the primary contributions to the first two phases (wind-up, stride), which are about creating a stable platform for the development of the forces required to propel the ball, and the initial stages of the kinetic chain moving those forces towards the throwing arm. The arm then becomes the primary element of the next three phases (arm cocking, acceleration, and deceleration); the final phase, follow-through, brings the rest of the body back to the fore [49]. There is therefore at least some case to be made that just analysing the behaviour of the arm is legitimate. Of course, manipulating target distance means changing the forces required to cross the space, which means the changing target affordance could also be predicted to affect the initial phases. So while isolating how the target affordances affect the arm phases is an option, it will be worth investigating all of this explicitly in the future. We have focused here on how an affordance analysis can constrain the selection of performance variables, but this issue reinforces the point made above that it can (and should) also constrain the selection of elemental variables.

### Throw standardisation

One basic assumption of the analysis is that each of the 20 trials that went into each analysis was an example of the person attempting to do the same thing; that there is in fact 'repetition', even if that happens 'without repetition'. We have explicitly looked for this in the shoulder-elbow-wrist parts of the throwing action, but if there is not enough 'repetition' in the components we haven't analysed, this could confound our analysis. Experimentally, throwing research often tries to solve this by literally isolating the arm elements of the throwing action by restraining the torso [e.g. 32]. We worked to standardise what people were doing via constraints; the verbal task instruction to start from the same location (centred with respect to the target) and to take a single step to produce the throw. We are confident that this was a sufficient constraint, but we just wanted to highlight that this is an important consideration for studying more complicated movements.

## Summary

The ecological approach proposes that the effectivity-calibrated perception of affordances is the mechanism for both action selection and control. Organisms perceive the affordances on offer, and this provides the key constraint for the soft assembly of a redundant action system into a task-specific synergy that can produce stable and functional behaviour; which dynamical effectivity properties are required are picked out by the dynamical affordance properties the effectivity dynamics must complement. In UCM terms, affordances guide the organism's selection of the set of elemental variables as well as performance variables, and a careful ecological analysis grounded in the task affordances to identify the complementary effectivities can therefore also constrain the researcher's selection of these variables. We have not fully connected these dots in this paper, but we have taken an initial step by explicitly including affordances, shown that it is viable, and seen clues in the data about the role effectivity dynamics also play in action control. Future work can measure or manipulate throwing effectivities; anything that affects the production of the release parameters from the affordance analysis, for example. There remains much to do, but the field now has many powerful analysis methods that embrace motor abundance, and the ecological approach provides the necessary theory and hypotheses those methods can be used to test.

## Acknowledgments

We would like to thank Harjiv Singh, Michael Collins, and Athanassios Bissas for their help with data collection and access to lab resources. We would also like to thank Raoul Bongers and his lab group for their extensive and invaluable feedback on drafts of this paper.

## Author Contributions

**Conceptualization:** Andrew D. Wilson.

**Data curation:** Timothy Bennett, Liam Thomas, Andrew D. Wilson.

**Formal analysis:** Timothy Bennett, Liam Thomas, Andrew D. Wilson.

**Investigation:** Timothy Bennett, Liam Thomas, Andrew D. Wilson.

**Methodology:** Timothy Bennett, Liam Thomas, Andrew D. Wilson.

**Project administration:** Andrew D. Wilson.

**Software:** Timothy Bennett, Liam Thomas.

**Writing – original draft:** Andrew D. Wilson.

**Writing – review & editing:** Timothy Bennett, Liam Thomas, Andrew D. Wilson.

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
